# Video-rate 3D imaging of living cells using Fourier view-channel-depth light field microscopy

Chengqiang Yi[1,5], Lanxin Zhu[1,5], Jiahao Sun[1], Zhaofei Wang[1], Meng Zhang[1,2], Fenghe Zhong[1], Luxin Yan[3], Jiang Tang[1], Liang Huang[4], Yu-Hui Zhang[2], Dongyu Li [1✉] & Peng Fei [1]

Interrogation of subcellular biological dynamics occurring in a living cell often requires noninvasive imaging of the fragile cell with high spatiotemporal resolution across all three dimensions. It thereby poses big challenges to modern fluorescence microscopy implementations because the limited photon budget in a live-cell imaging task makes the achievable performance of conventional microscopy approaches compromise between their spatial resolution, volumetric imaging speed, and phototoxicity. Here, we incorporate a two-stage view-channel-depth (VCD) deep-learning reconstruction strategy with a Fourier light-field microscope based on diffractive optical element to realize fast 3D super-resolution reconstructions of intracellular dynamics from single diffraction-limited 2D light-filed measurements. This VCD-enabled Fourier light-filed imaging approach (F-VCD), achieves video-rate (50 volumes per second) 3D imaging of intracellular dynamics at a high spatiotemporal resolution of ~180 nm × 180 nm × 400 nm and strong noise-resistant capability, with which light field images with a signal-to-noise ratio (SNR) down to -1.62 dB could be well reconstructed. With this approach, we successfully demonstrate the 4D imaging of intracellular organelle dynamics, e.g., mitochondria fission and fusion, with ~5000 times of observation.

[1] School of Optical and Electronic Information-Wuhan National Laboratory for Optoelectronics-Advanced Biomedical Imaging Facility, Huazhong University of Science and Technology, Wuhan 430074, China. [2] Britton Chance Center for Biomedical Photonics-MoE Key Laboratory for Biomedical Photonics, Advanced Biomedical Imaging Facility-Wuhan National Laboratory for Optoelectronics, Huazhong University of Science and Technology, Wuhan, Hubei 430074, China. [3] State Education Commission Key Laboratory for Image Processing and Intelligent Control, Huazhong University of Science and Technology, Wuhan, Hubei 430074, China. [4] Department of Hematology, Tongji Hospital, Tongji Medical College, Huazhong University of Science and Technology, Wuhan, Hubei 430030, China. [5] These authors contributed equally: Chengqiang Yi, Lanxin Zhu. ✉email: li_dongyu@hust.edu.cn

Unraveling the spatiotemporal regulation and dynamics inside living cells is crucial for advancing fundamental biological research[1–4]. Many fast biological processes occur in three dimensions at a submicron scale and successively change across a long timescale, thereby posing a big challenge for current fluorescence microscopy techniques. While scanning-based light sheet microscopy[5,6], confocal microscopy, and structure illumination microscopy[7] can provide high spatial resolution in three-dimensional space, they all suffer from compromised volumetric imaging speed and strong photodamage owing to the repetitive light exposure to the samples.

The emerging light-field microscopy (LFM) delivers superior temporal resolution and photon efficiency through encoding both spatial and angular information of 3D signals into a single 2D camera snapshot, thus yielding 3D reconstruction without scanning[8]. However, the poor and nonuniform spatial resolution, as well as the presence of reconstruction artifacts[8,9], greatly limit its application in subcellular imaging beyond the diffraction limit[10].

Unlike standard LFM modulating the light at the native image plane, its structure variant Fourier light field microscopy (FLFM) records the spatial and angular information in the Fourier domain, permitting spatially invariant sampling and leading to a significant reduction of sampling-induced artifacts[11]. In addition, FLFM overcomes the spatial sampling limitation of standard LFM and leverages the optical aperture to achieve improved spatial resolution[11–13].

Despite the abovementioned advances, the photon budget constraint still exists in FLFM. The spatial resolution remains sacrificed to obtain the angular information, making the subcellular structures difficult to distinguish. When the signal turns weak after long-term excitation and fluorescing, undesirable ringing artifacts also arise[14,15] in the reconstructed results, further diminishing the applicability of FLFM in live-cell imaging. Super-resolution radial fluctuations (SRRF) algorithm has been applied to light-field views to obtain high-resolution 3D reconstruction of FLFM beyond diffraction limit[16]. However, the super-resolution images generated by SRRF usually suffer from severe artifacts, especially under suboptimal imaging conditions with low signal-to-noise ratio (SNR) or insufficient sampling[17,18]. Also, the fabrication of current Fourier microlens arrays is challenging, owing to the requirement of large pitch sizes and small surface curvatures, leading to a high cost.

Deep neural networks (DNNs) have emerged as a powerful tool to overcome the limitations imposed by photon budgets for their strong fitting ability and their capacity to incorporate abundant prior knowledge[19–26]. Our previous development of view-channel-depth (VCD) deep-learning strategy successfully exceeds the photon budget limitation of conventional LFM and achieves 3D reconstructions from single 2D LFM snapshots[27–30]. Given that severely under-sampled spatial-angular information coupled with varying noise in the light-field imaging of live cells, super-resolution 3D reconstruction from a single 2D raw light-field image (LF) remains highly challenging for standard VCD-based light-field imaging.

Here, we report a FLFM system based on the diffractive optical element (DOE). Besides the low cost and ultracompact structure, the customized DOE Fourier lens can maximize the modulation of incident light and overcome the obstacles in the fabrication of conventional Fourier microlens. Moreover, inspired by the progressive image enhancement[19], we present a two-stage VCD reconstruction strategy, F-VCD, which enables fast, long-term 3D imaging of live cells at sub-200 nm resolution. F-VCD decomposes complex FLFM reconstruction problems into two procedures: view-correlated denoising and limited-view 3D reconstruction, which narrows the gap between the corrupted 2D LFs and high-quality 3D stacks. Unlike the conventional VCD model, it contains an additional view-attention network to denoise the light-field (LF) views with various levels of noise. To reconstruct 3D information (over 40 slices) from only 3 FLFM

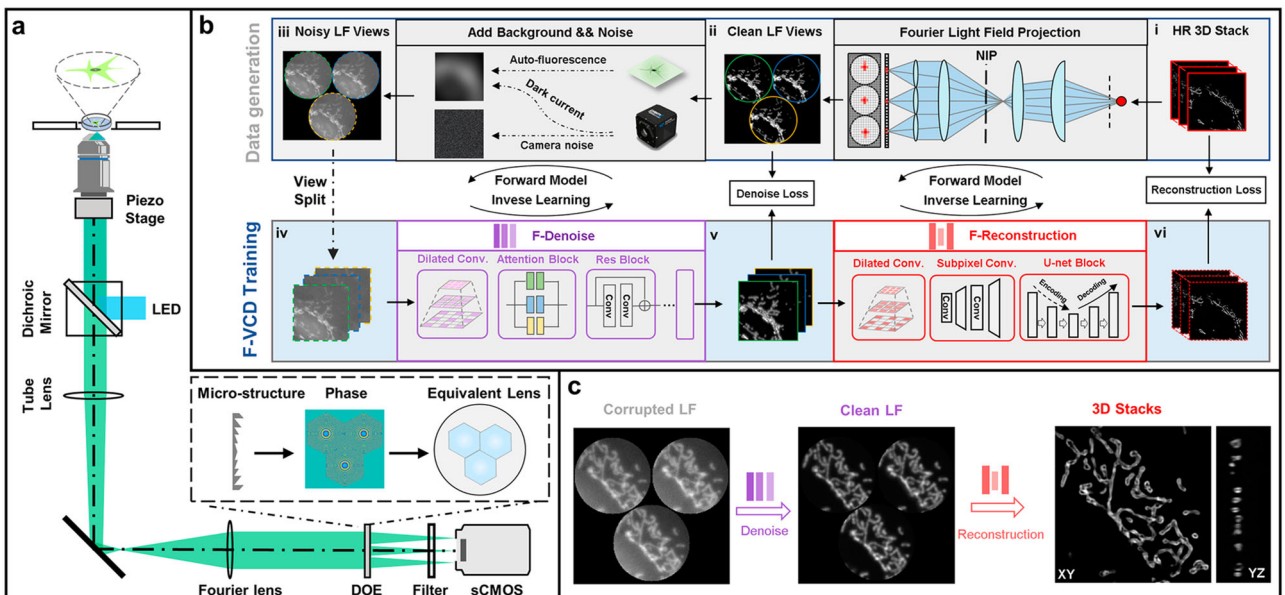

**Fig. 1 The principle of F-VCD. a** The optical setup of the FLFM system. The inset shows the phase structure of the DOE and its equivalent MLA. **b** The pipeline of F-VCD construction, including data generation (the upper row) and F-VCD training (the bottom row). In the data generation branch, via simulation, high-resolution stacks are first transformed to light-field views, and then these views are degraded by noise and background quantified from experiments to yield "Noisy LF views". In the F-VCD training branch, these noisy views are sent to "F-Denoise" network to suppress the severe noise. Finally, with the F-Reconstruction network, high-resolution 3D results can be produced. Among the network inference, each network is optimized by the corresponding loss function (denoise loss for F-Denoise, reconstruction loss for F-Reconstruction). **c** Real-time inference of 3D images from the recorded 2D light-field images by a trained F-VCD model.

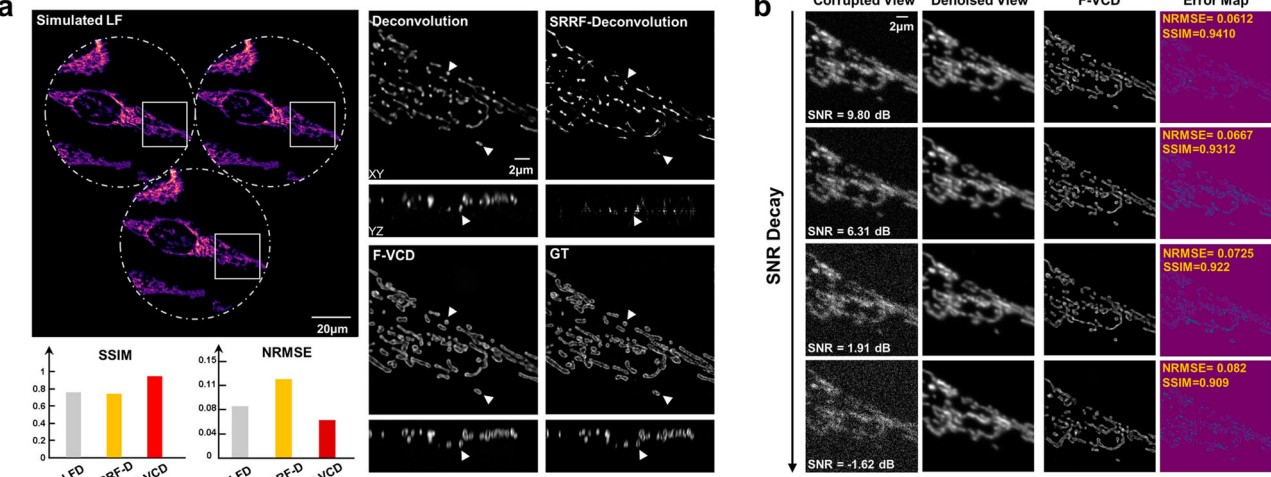

**Fig. 2 Comparative performances of FLFM deconvolution, SRRF-deconvolution, and F-VCD on the reconstruction of mitochondria. a** Comparison between deconvolution-based reconstruction and F-VCD reconstruction. The white arrowheads indicate the higher contrast and fidelity of F-VCD SSIM and NRMSE metrics, which were calculated to quantitatively evaluate the reconstruction accuracy of F-VCD. **b** The performance of F-VCD under different SNR. Error maps were calculated between ground truths and different F-VCD results.

views, triple dilated convolution layers with different kernels were introduced into F-VCD to extract and aggregate multi-channel features from FLFM views, substantially augmenting the input information for the F-VCD model. With F-VCD, we successfully demonstrate 3D super-resolution imaging of live cells at ~180 nm × 180 nm × 400 nm resolution across ~81 µm × 81 µm × 7 µm volume with a volume rate up to 50 Hz and maintain a high reconstruction quality even when the SNR of LF images becomes very low. We also capture the fast locomotion and morphology changes of mitochondria over 4 minutes, containing about 5000-time points of observation. Diverse mitochondrial fusion, fission, and mitochondrial dynamic tubulation (MDT) phenomena have been identified and quantitatively analyzed based on the F-VCD live imaging results.

## Results

**Principle of F-VCD strategy**. We built a DOE-based Fourier light field microscopy to catch the fast 3D dynamics as shown in Fig. 1a. The transmittance function of the DOE was designed to be similar to a previously proposed conventional microlens array[12], but the price was dozens of times lower because DOE was much easier to fabricate. To solve the complex task of denoising and super-resolution 3D reconstruction in such light-field imaging of live cells, we developed a two-stage network, F-VCD, to subsequently conquer two outstanding problems of denoising and 2D-to-3D transformation. The principle of F-VCD is shown in Fig. 1b, and its detailed architecture is shown in Supplementary Fig. 1. Before model training, dual-stage ground truths (GTs) (HR Stacks and Clean LFs) and synthetic light-field images (Corrupted LFs) are generated according to the following physics-modeled processes: (i) The "HR Stacks", which are obtained from a commercial 3D microscope (ZEISS LSM 980 with Airyscan 2), serve as the GTs for the reconstruction module and are resampled to match the sampling rate of FLFM setup; (ii) The "Clean LFs" are designed to guide the denoising task and generated by the convolution of 'HR Stacks' with Fourier light field point spread function based on wave-optics theory; (iii) The "Corrupted LFs" are generated by adding Gaussian noise, Poisson noise, and specific light-field background, to the "Clean LFs" based on the properties of diverse measured LFs.

We specifically designed two network modules to progressively denoise the corrupted Fourier LFs and reconstruct the denoised LFs.

The first module is a view-correlated denoise module, termed "F-denoise" module, which converses the noised views (Fig. 1b-iv) into clean ones (Fig. 1b-v). The presence of parallax and variations in intensity distribution among the three views results in differences in SNR and spatial resolution, highlighting the importance of assigning different weights to each of the views for optimal denoising results[14,31]. To mitigate these discrepancies, we introduced a view-attention branch in a conventional RCAN network, enabling a balanced influence on network inference results. The second module of F-VCD, F-Reconstruction, functionalized as 'Fourier light-field reconstruction', is a "limited-view to 3D depth" transformation module to reconstruct 3D volumes (41slices, Fig. 1b-v) from a few views of FLFM (three views in our case, Fig. 1b-iv). Unlike conventional LFM encoded with hundreds of views, 3D reconstruction from extremely few views in FLFM is challenging for previous VCD network[27]. Therefore, we added dilated convolutional layers to extract multi-scale features before VCD to increase the output channels extracted from the three views. Furthermore, the losses in these two modules are weighted to preserve the continuity of data distribution when coupling two pseudo-inverse learning. More details about F-VCD implementation are described in Supplementary Fig. 1 and the "Methods" section.

By iteratively minimizing the weighted loss with the guidance of dual-step GT, F-VCD can be well trained to instantly provide noise-free and high-resolution 3D reconstruction from noisy 2D LF images captured from a lab-built FLFM system (Fig. 1c). Compare with the low-speed traditional deconvolution-based reconstruction method, F-VCD achieves hundreds of times faster reconstruction (4–13 s vs. 77 ms, Supplementary Fig. 2).

**F-VCD enables high-fidelity reconstruction under various SNRs.** We first tested the reconstruction resolution of F-VCD and its fidelity using simulation data. As shown in Fig. 2a, the outer mitochondrial membrane could not be resolved at all by the deconvolution algorithm. With the one-frame SRRF strategy[16], although the resolution seemed to be improved, it merely made the mitochondria appear thinner but failed to distinguish the outer membrane. In addition, there were notable artifacts shown in the axial image. Therefore, its structural similarity index (SSIM) was even lower than that of the deconvolution algorithm. In contrast, F-VCD remarkably enhanced the reconstruction quality of FLFM, with which the membrane structure of the mitochondria was

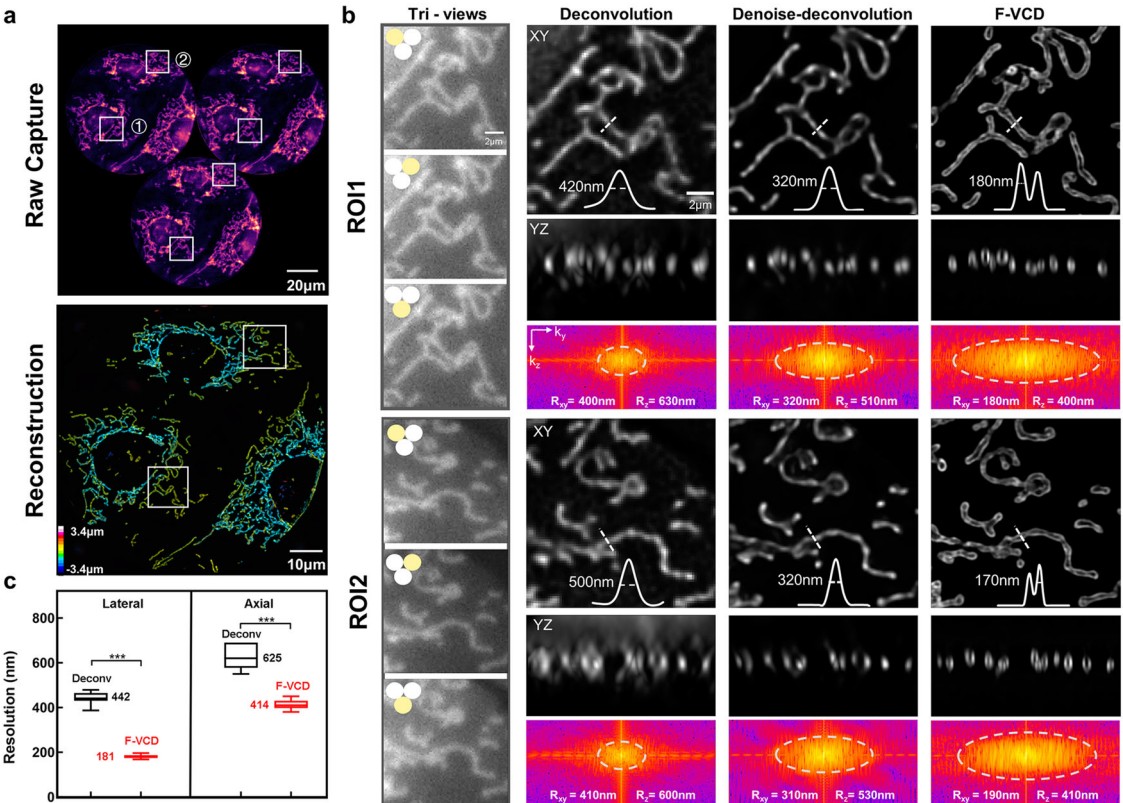

**Fig. 3 Resolution enhancement on fixed mitochondria using F-VCD. a** Top: raw FLFM views with noises; Bottom: F-VCD reconstruction. **b** magnified views of mitochondria (ROI 1 and 2 marked with white boxes in (**a**)) and their corresponding 3D reconstructions by conventional deconvolution, denoised deconvolution and F-VCD, respectively. The comparative frequency spectrums indicate that F-VCD reconstructs mitochondria with the highest spatial resolution. **c** Resolution quantification from frequency analysis on fixed mitochondria. Lateral and axial values are shown for deconvolution reconstruction (Lateral: 442.3 ± 29.2 nm, Axial: 625 ± 54.6 nm, respectively; p-value: $3.4016 \times 10^{-11}$) and F-VCD reconstruction (Lateral: 181.3 ± 10.0 nm, Axial: 414 ± 21.6 nm, respectively; $p$-value: $1.152 \times 10^{-7}$).

clearly reconstructed with higher contrast in both the x–y and x–z planes. Meanwhile, the F-VCD result showed a very high SSIM of 0.941, as compared to the values lower than 0.8 in SRRF and deconvolution results. The normalized root mean square error (NRMSE) values also indicated that F-VCD had the best reconstruction accuracy among the three reconstruction approaches.

Then we tested the robustness of F-VCD under different SNRs on the simulation data to evaluate its capability of monitoring the living cells in the long term. As shown in Fig. 2b, F-denoise could recover the image even with an ultra-low SNR down to −1.62 dB. By using the whole F-VCD, 3D super-resolution reconstruction was realized with a high structure fidelity of 0.91 SSIM.

**High-resolution reconstruction of fixed cells using F-VCD.** We assessed the performance of F-VCD on fixed U2OS cells expressing Tomm20-EGFP-labeled outer mitochondrial membrane signals. F-VCD realized 3D reconstruction with the hollow membrane structure clearly discernable (Fig. 3a). Statistical analysis revealed that F-denoise could reduce the axial artifacts and slightly improve the resolution from ~400 nm to ~320 nm for conventional FLFM deconvolution, and F-VCD further provided 2× lateral and 1.5× axial resolution improvement with minimal artifacts (Fig. 3b, c). Compared to U-net and RCAN, F-denoise showed superior performance on denoising FLFM views and extending equivalent resolution and frequency spectrum (Supplementary Fig. 3). Together with the limited-view reconstruction module (F-Reconstruction), this progressive optimization strategy achieved higher 3D resolution, fidelity, and fewer artifacts than one-stage reconstruction network (Supplementary Fig. 4). Besides, compared with

original VCD, F-Reconstruction shows higher spatial resolution and contrast by the extra multi-scale features extraction module (Supplementary Fig. 5). We further evaluated the ability of F-VCD reconstruction using U2OS cells expressing EGFP-Sec61β-labeled endoplasmic reticulum (ER) signals. Despite the high-level background and densely-labeled signals, F-VCD accurately predicted the 3D distribution of the ER (Fig. 4). In contrast, deconvolution reconstruction suffered from poor resolution, excessive axial artifacts, and noticeable signal loss. It should also be noted that F-VCD successfully reconstructed an axial range of ~7 μm, surpassing the depth-of-field (DoF) limitation imposed by the optical hardware and enabling complete visualization of an entire cell.

**4D visualization and quantitative analysis of mitochondrial dynamics in a living cell.** With F-VCD-enhanced FLFM, we demonstrated video-rate 4D observation of mitochondrial dynamics in a living cell for the long term. A low excitation power down to 0.22 mW (focal plane of the objective) was set to minimize the phototoxicity and unsurprisingly led to limited SNR and reduced effective image bandwidth in the captured views. Despite these low-quality inputs, the F-VCD network successfully denoised the views and provided high-resolution 3D reconstructions, as shown in Fig. 5a. It is noted that, during the consecutive imaging, the signals were also bleached gradually. This phenomenon is inevitable for live-cell imaging and is evidenced by the decreased cut-off frequency in the raw captured LFs. The quality of deconvolution results degraded quickly as a consequence of such signal bleaching (Fig. 5b, Supplementary Fig. 6). In sharp contrast, F-VCD maintained the resolutions of

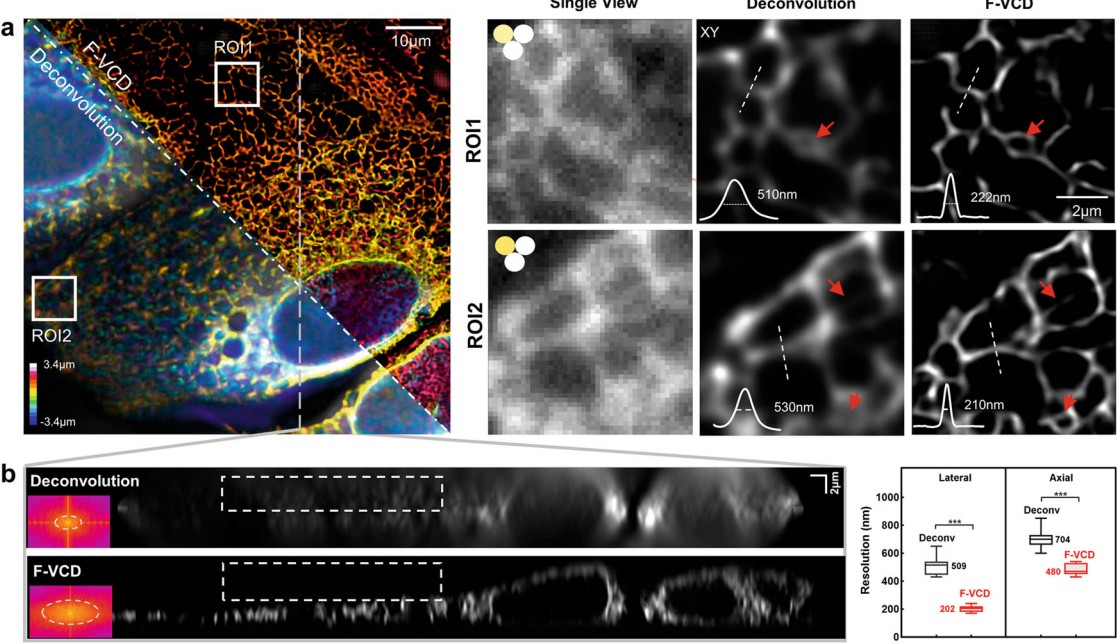

**Fig. 4 3D imaging of fixed endoplasmic reticulum with F-VCD. a** Comparison of deconvolution-based reconstruction and F-VCD reconstruction. The magnified images of two selected ROIs show a significant quality difference between the two reconstruction approaches. **b** The y–z section plane of deconvolution results and F-VCD results along the white dash line in (**a**). The frequency spectrums of the axial planes show the spatial resolution enhancement offered by F-VCD. The white frames indicate that F-VCD outperforms in artifact suppression. The box plots show the statistics results of resolution quantification of the deconvolution approach (lateral: 509.0 ± 67.5 nm, axial: 704.6 ± 70.2 nm; $p$-value: 2.82 × 10$^{-8}$) and F-VCD (lateral: 202.0 ± 23.4 nm, axial: 480.9 ± 37.2 nm; $p$-value: 9.99 × 10$^{-4}$).

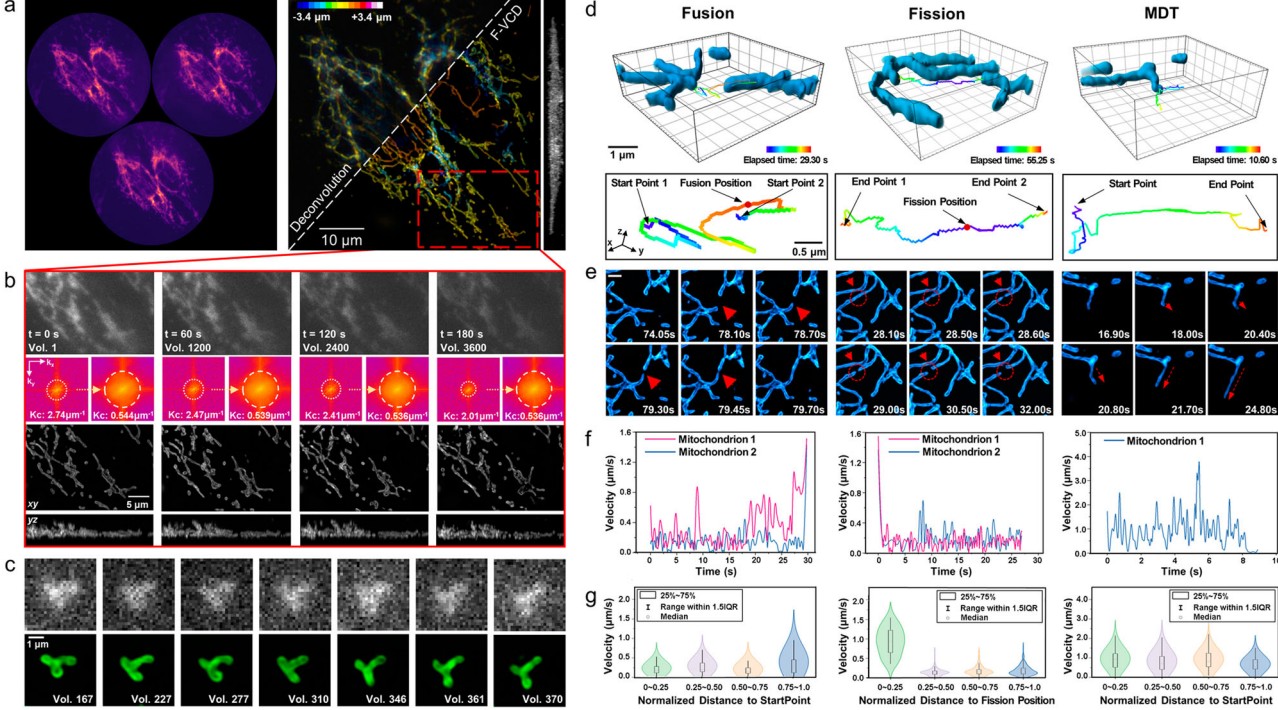

**Fig. 5 Imaging and analyzing mitochondrial dynamics in living cells using F-VCD enhanced FLFM. a** Left: raw LF image at $T = 0$ s; Right: comparative deconvolution and F-VCD reconstructions of captured LF. **b** The magnified views of the red rectangular region shown in (**a**), demonstrate the enhancement of F-VCD across the entire acquisition time of 3 min. **c** Fast morphological dynamics of mitochondria. Top panel: captured LF views; Bottom panel: the MIPs of F-VCD reconstruction. **d** 3D tracking of mitochondrial fusion, fission, and tubulation. **e** Representative images related to the three types of mitochondria events shown in (**d**). Scale bar:1 μm. **f** Velocity changes during the occurrence of the three mitochondrial events. **g** Mitochondrial velocity as a function of distance during the occurrence of the three events.

reconstructed images across ~5000 times of 3D observation (Fig. 5b, Supplementary Video 1), indicating the better robustness of F-VCD under low photon budget, compared with scanning-based imaging modalities (less than 50 volumes, Supplementary Fig. 7). With robust performance on various FLFM inputs, F-VCD achieved fast and sustained 3D observation of mitochondrial dynamics, thereby allowing quantitative analysis of mitochondrial fusion, fission and MDT at milli-second temporal resolution and 180 nm spatial resolution (Fig. 5c, e, Supplementary Fig. 8, Supplementary Video 2). We performed 3D tracking of the mitochondria during the abovementioned events (Fig. 5d) to analyze their velocity dynamics (Fig. 5f, g). Our results revealed a distinct velocity jump occurred during mitochondrial fusion and fission, in contrast to the relatively constant velocity during MDT. It was observed that mitochondrial velocity tended to be significantly higher at the end of fusion and at the beginning of fission. For the quantitative validation of F-VCD's performance, we conducted an in situ comparison between traditional imaging methods (confocal and wide-field imaging, Supplementary Fig. 9) and F-VCD reconstruction both on fixed and living cells, and Squirrel analysis[32] was performed. The results showed high structural similarity and few reconstruction errors of F-VCD, indicating that the proposed imaging strategy could be used to perform biological downstream analysis.

## Conclusion and discussion

High spatial resolution, high volumetric imaging speed, and low light exposure have to be carefully balanced in the current fluorescence imaging of live cells. F-VCD surpasses this limitation by incorporating an improved VCD model with DOE-based Fourier light field microscopy, achieving video-rate, super-resolution 3D reconstructions of intracellular dynamics from single diffraction-limited 2D light-field measurements.

In terms of light-field encoding, the use of DOE in self-built FLFM dramatically reduces the cost of the system and allows readily accessible light-field modulation. Since DOE has the capability of flexible phase modulation, the combination with multi-focus imaging[33,34] or enhanced DoF modulation[35] might further push the limits of FLFM. In terms of light-filed reconstruction, our F-VCD shows ~2.2-times improved resolution accompanied by ~30% increased reconstruction accuracy, as compared to the conventional deconvolution-based FLFM approach Also, unlike regular VCD that directly predicts depth information from corrupted views, F-VCD follow a divide-and-conquer strategy to fulfill the coupled denoise and view-to-depth transformation tasks, thereby showing notably improved performance when dealing with live-cell imaging with various conditions. Besides, with the fast inference ability of the network, F-VCD achieves hundreds of volumetric reconstruction accelerations compared with deconvolution (77 ms vs. 4–13 s). This computation time could be even further reduced with multi-GPU parallel computing or model acceleration[36] in future work. It is worth mentioning that, as a view-wise attention-denoising strategy, which is independent of the arrangement mode of input views, F-Denoise could also be used to improve the quality of each view and reconstruction of conventional LFM (Supplementary Fig. 10).

F-VCD demonstrates its strong capability of imaging the fast morphological changes of mitochondria, including MDT, fission, and fusion, and the follow-up imaging-based quantitative analysis. We believe our proposed imaging strategy would be a valuable and readily accessible tool for structural studies of cell biology. Here, we only demonstrated the capability of F-VCD on living cell imaging under the high-magnification-imaging hardware setting, which led to a limited DOF (~7 μm). We anticipate expanding the utility of F-VCD to other biological research that requires fast and clear observations, like in vivo imaging, by designing suitable hardware settings.

## Methods

**DOE-based FLFM optical setup.** The FLFM system is based on a commercial inverted microscope (Olympus IX73) (Fig. 1a). A 100 × /1.4 NA objective (Olympus UPlanSApo 100 × /1.4) is used to collect fluorescence signals and is controlled by an objective scanner (P73.Z200s, Coremorrow) for foci planes shift. An iris (SM1D12, Thorlabs) is positioned at the native image plane (NIP) to adjust the field of view of imaging in order to avoid view-overlapping. A Fourier lens (FL, ACT508-300 Thorlabs) is used to make optical Fourier transformation. To maximize the utilization of the full Fourier aperture and make the optimum balance between resolution and FOV, we customize a low-cost silica DOE (ZheYan Technology) to function as a Fourier MLA (pitch $d = 3.25$ mm, f-number = 37, $f_{ML} = 120$ mm) as illustrated in the inset of Fig. 1a. The DOE is placed at the back focal plane of the FL, which is conjugated to the back pupil of the objective to segment the Fourier pupil into three views. The light field signals are then captured by a sCMOS (Prime BSI Express, Teledyne Photometrics) located at the back focal plane.

**Architecture of F-VCD neural network.** F-VCD is composed of two modules: "F-Denoise" and "F-Reconstruction" (see Supplementary Fig. 1). In the F-Denoise module, we adopt the well-known RCAN architecture for feature extraction but add another "view-attention" branch to match the different resolution or SNR in triple views. Due to the dense connection among residual channel attention blocks of RCAN, the low-spatial-frequency information can be easily bypassed among networks, which is beneficial to the prediction of high-spatial-frequency information. In addition to the original channel attention branch, we adjust the data dimension (view,h,w,c) to (c,h,w, view) so that the attention block can extract the view-wise coefficient to reweight the extracted feature. In the F-Reconstruction module, based on our previous VCD network[27], we add three dilation convolution blocks to increase the number of input channels (3 for designed triple-views LF) and upscale the lateral size of the extracted feature. We also modify the original encoding block of U-Net in VCD by replacing the plain convolution operation with a residual block for network convergence. The normalization layer and activation function are also changed into Instance Normalization and LeakyRelu to avoid the truncation of weak signals in the optimization of the deep network.

**Data preprocessing.** In the imaging process of Fourier light field microscopy (FLFM), the captured LF termed $y$ can be formulated as:

$$y = H * x + n + b$$

where $x$ is the original 3D signal of the sample and $H$ is the point spread function of FLFM. Due to the noise ($n$) and background ($b$) from the camera's dark current, deconvolution-based algorithms produce ringing artifacts and miscalculated signals. To tackle the inverse problem of reconstructing the true signal ($x$) from its degraded observation ($y$), F-VCD employs a two-step approach that addresses each subproblem in turn: denoising and 3D reconstruction. So, the training dataset contains clean LF, noisy LF, and corresponding high-resolution 3D stack. We use a scalar diffraction model based on the Fresnel diffraction formula to calculate the PSF of FLFM. Synthetic LF projections were generated by convolving of PSF and 3D stacks. According to the measurement of experimental data such as SNR and signal-to-background ratio (SBR), corresponding noise (Poisson noise and

Gaussian noise) and constant background value were added into synthetic LFs. Besides, due to the requirement of the VCD network input format, triple views need to be extracted from LF projections according to the location and pitch of each microlens. Finally, for data augmentation and memory limitation, random image flip, rotation, and crop were operated to establish a training data set.

**Training details of F-VCD**. In the training stage, the F-VCD network is updated by optimizing two sub-nets simultaneously. For denoise net (F-Denoise), we use weighted L1-L2 loss as a loss function in consideration of the denoise ability of L1 loss and convex properties of L2 loss. For reconstruction net (F-Reconstruction), L2 loss and gradient loss are both used to guarantee pixel-wise regression and high-frequency details recovery. We chose an Adam optimizer with a decay learning rate and weighted 'denoise-reconstruction' loss function (e.g., 0.2 and 0.8, respectively). In addition, pretraining on denoise net and reconstruction net separately helps the quick convergence of joint optimization, and we recommend enlarging the lateral size of pretraining data for better performance. For example, $160 \times 160 \times 3$ for pretraining denoise net while $80 \times 80 \times 3$ for joint optimization.

After the convergence of F-VCD, the model would automatically save the best parameters for inference. Before 3D reconstruction by inference, experimental light-field data were realigned and referred to as synthetic projection. A convenient way is to capture an LF image full with a constant value or signal to get the rotation degree and microlens position. Once realigned, view stacks can be obtained by cropping triple views from the input experimental light fields.

**Sample preparation**. U2OS cells were grown in culture medium containing McCoy's 5 A medium (Thermo Fisher Scientific) supplemented with 1% antibiotic-antimycotic (Thermo Fisher Scientific) and 10% fetal bovine serum (Thermo Fisher Scientific) at 37 °C with 5% $CO_2$ in a humidified incubator. For labeling mitochondria or ER in fixed and living cells, U2OS cells were first transfected with Tomm20-EGFP for mitochondria and Sec61β-EGFP for ER using Lipofectamine 2000 according to the standard protocol and cultured at 37 °C with 5% $CO_2$ for an additional 24 h. For fixed cell imaging, the cells were fixed with 2% glutaraldehyde for 20 min.

**Quantification of SNR**. The SNR was calculated by:

$$\text{SNR} = 10 \cdot \lg \left( \sum_{x=0}^{M-1} \sum_{y=0}^{N-1} [f(x,y) - \bar{f}]^2 \Big/ \sum_{x=0}^{M-1} \sum_{y=0}^{N-1} [\hat{f}(x,y) - f(x,y]^2 \right)$$

$$\bar{f} = \frac{1}{MN} \sum_{x=0}^{M-1} \sum_{y=0}^{N-1} f(x,y)$$

where $f(x,y)$ is the signal distribution and $\hat{f}(x,y)$ is the noisy image.

**Statistics and reproducibility**. We randomly split all datasets into training data, validation data, and test data to check the model performance across the training phase. The networks were trained and tested multiple times for 3D reconstruction of mitochondria and ER data to find the optimal set of hyperparameters. The number of training datasets was chosen by the quality of prediction results. The statistical analysis and plotting were completed in GraphPad Prism. Replicates were defined as images obtained from different field-of-views. For fixed and living cell imaging, data was collected under different visits to ensure the reproducibility of our model.

**Reporting summary**. Further information on research design is available in the Nature Portfolio Reporting Summary linked to this article.

## Data availability

All source data for all graphs and charts in the main and supplementary figures can be found in Supplementary Data.

## Code availability

F-VCD source code and example data have been uploaded to Github.

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

## Acknowledgements

We thank the funding support by the National Key Research and Development Program of China (2022YFC3401102 and 2017YFA0700501), the National Natural Science Foundation of China (T2225014, 21874052, 61860206009, and 62375096) and Hubei Provincial Natural Science Foundation of China (2023AFB861).

## Author contributions

C.Y. was involved in the experiment setup, investigation, statistical analysis, and writing. L.Z. was involved in the experiment setup, investigation, writing, and editing. J.S. and Z.W. were involved in image processing and statistical analysis. M.Z. was involved in the sample preparation. F.Z., L.Y., J.T., L.H., and Y.H.Z. were involved in writing and editing. D.L. was involved in conceptualization, writing, and editing. P.F. was involved in the conceptualization, writing, editing, and project management.

## Competing interests

The authors declare no competing interests.
