## [Peer Review File · Communications Biology]

Reviewers' comments:

Reviewer #1 (Remarks to the Author):

The authors proposed a new network termed F-VCD to achieve high-quality reconstruction for FLFM. F-VCD incorporates a two-stage strategy for both denoising and deconvolution tasks. Overall, I think it is a nice work for high-performance 3D living cell imaging using FLFM. I would recommend its publication in *Communication Biology* if the following points are addressed.

1. In terms of hardware, Shu Jia et al [12] have proposed a similarly shaped microlens array to achieve high-resolution FLFM imaging in living cells. But in the experiments of this paper, the proposed DOE seems to play exactly the same role as the microlens array in [12]. Therefore, this DOE does not seem to be of significance. However, I don't think the DOE is very crucial for this paper.
2. In terms of the algorithm, the authors proposed an additional supervised denoising network. A natural question is whether the F-denoise is compatible with both FLFM and LFM data. Can people who only have conventional LFM use F-denoise for image denoising?
3. In Supplementary Fig. 3, F-VCD shows better performance than VCD and the authors attribute it to the limited-view reconstruction module. An ablation experiment on the module is necessary.
4. In the legend of Fig. 1c, the authors said "real-time inference". But I don't seem to find any relevant description of the inference time of the two networks.
5. Several limitations should be involved in the discussion section, e.g., the shallow depth of field, inapplicability to in vivo imaging, etc.
6. It is recommended that the relevant code be made publicly available rather than requested from the authors so that the broad readers of *Communication Biology* can easily verify it.
7. Missing reference for other learning-based LFM methods, such as:
Lu, Z., Liu, Y., Jin, M. et al. Virtual-scanning light-field microscopy for robust snapshot high-resolution volumetric imaging. *Nat Methods* 20, 735–746 (2023). <https://doi.org/10.1038/s41592-023-01839-6>

Reviewer #2 (Remarks to the Author):

In this article, the authors combine a two-stage view-channel-depth (VCD) deep learning reconstruction strategy with a Fourier light field microscope based on a customized diffractive optical element to achieve a fast 3D super-resolution reconstruction of intracellular dynamics measured by a single diffractive limited two-dimensional light field. This Fourier light field imaging method was claimed to improve 3D imaging of intracellular dynamics at video rate (50 volume/second), with high spatiotemporal resolution of $\sim 180 \text{ nm} \times 180 \text{ nm} \times 400 \text{ nm}$ and strong anti-noise capability, and can well reconstruct light field images with a low signal-to-noise ratio (SNR). The authors demonstrated 3D imaging of mitochondrial fission and fusion in the cells. The research has an innovative basis and the results are satisfying. However, there are some issues to be solved in order to be published.

1. "We specifically designed two network modules to progressively denoise the corrupted Fourier LFs and reconstruct the denoised LFs. The first module is a view correlated denoise module, termed 'F-denoise' module, which converses the noised views into clean ones. .. The second module of F-VCD,

functionalized as 'Fourier127 light-field reconstruction', is a 'limited-view to 3D depth' transformation module to reconstruct 3D volumes...". The authors should clearly explain the purpose of doing this, i.e., what is the necessity and significance of the noise reduction image obtained in the first stage reconstruction before the 3D reconstruction in the second stage? In my opinion, the 3D reconstruction already includes the denoising result, why not merge it and get 3D volumes in one step? The reported deep networks have the ability to do this.

2. "As shown in Fig. 2a, ... In contrast, F-VCD remarkably enhanced the spatial resolution of FLFM ...". Here I did not see the resolution quantification in Fig. 2. I think the authors meant that the F-VCD remarkably enhanced the image contrast which also corresponds to clear detailed structure and is easy to confused with resolution. The spatial resolution was measured and presented in the next figures but not here.

3. The authors mentioned the resolution improvements several times, but there is no statistical analysis of this, only some subjective selections of a profile across the mitochondria (Figs. 3 and 4). The authors should perform more PSF or profile measurements to verify the resolution improvement, and analyze their statistical parameters, such as mean, variance, p-value, etc.

4. The network reconstruction speed is an important metric because we do not want the reconstruction speed to be so slow exceeding the acquisition time of high-SNR images. The author should provide additional information in this regard.

5. The annotation and text in the figures are too small to read (especially Figures 1 and 5). The authors should realize that this is very important for reviews' and readers' reading experience. The invisible plots affect our judgment of the reliability of the results.

6. The authors claimed to achieve the video rate in 3D imaging of intracellular dynamics at high spatiotemporal resolution, but no time-related results were given, nor did they supplement the video.

7. A SNR of 1.51 is not dramatically low. The SNR calculation adopted in this work is an alternative definition of SNR is as the reciprocal of the coefficient of variation, i.e., the ratio of mean to standard deviation of a signal or measurement. Notice that such an alternative definition is only useful for variables that are always non-negative, and it is only an approximation. Since GT image is given, SNR can be more precisely calculated as the ratio of the power of signal to the power of background noise. This can be obtained by computing the ratio of summed squared magnitude of the signal to that of the noise (GT - signal). Broadly, it is defined in decibels.

Reviewer #3 (Remarks to the Author):

The authors present a technique to reduce the effect of limited photon budget on the reconstruction quality obtained using Fourier Light Field Microscopy, by employing a deep learning based reconstruction and incorporating a diffractive optical element in place of the commonly used microlens array. In particular, the authors show that it is sufficient to form three light-field projections to generate a 3D reconstruction of the sample using this technique.

Although the approach is interesting, publication in any form is premature at this point. I summarize my main comments below.

#1: The paper is riddled with spelling and grammatical mistakes. The paper needs to be properly proof-read before resubmission to any journal.

#2: The figure captions need to be more descriptive - it is very difficult to understand the figures at present

#3: On the same note, figure 1 is very confusing, in particular 1b.

#4: The text on the plots in figure 5 is impossible to read.

These issues makes it difficult to assess the scientific quality of the paper, and needs to be fixed before a proper review can be made. Nonetheless, I have some questions alos on the scientific side:

#5: The approach is motivated by the need for generating high-quality reconstructions with low photon budgets to enable live-cell imaging without phototoxicity. However, this aspect is in my view not properly explored. In particular, I miss a quantitative comparison of the number of photons/volume needed to perform comparative reconstructions in various imaging modalities.

#6: It is unclear, which of the improvements are due to the diffractive optical element, and which are due to the deep learning algorithm. This needs to be properly explained

#7: There is no quantitative validation of the deep neural network performance. This is extremely important in this type of work, seeing as the networks are trained to optimize some metric on a large dataset, which does not necessarily guarantee that the output can be used to draw downstream biological conclusions. Validating the biological conclusions on some test case where conventional imaging techniques can be used would be very valuable.

Are the claims novel?

Yes

Is the work convincing?

No (see point 7 above)

Will the paper influence thinking in the field?

No (the deployment of deep learning to enhance imaging is already happening at a fast pace, employing it on one more technique will not influence the thinking in the field)

In conclusion, I feel that any form of publication is premature at this point.

Re: Revision for “Communications Biology” Manuscript “**Video-rate 3D imaging of living cells using Fourier view-channel-depth (F-VCD) light field microscopy**” (# COMMSBIO-23-2128) authored by Chengqiang Yi, Lanxin Zhu, Dongyu Li (corresponding author) and Peng Fei.

We are grateful to the reviewers for their constructive and insightful comments that indeed help to improve this manuscript. Now we will respond to the comments one by one. **The responses to each comment are in blue.** *Quotations of comments from the Reviewers are in italic.* **Changes to the manuscript are in bold.** A **highlighted** version of the revised are included in the submission to mark these changes.

Sincerely yours,

Dongyu Li,

Associated Professor of School of Optical and Electronic Information,
Huazhong University of Science and Technology; Email: feipeng@hust.edu.cn

The responses to the reviewers' comments are listed below:

Reviewer #1:

The authors proposed a new network termed F-VCD to achieve high-quality reconstruction for FLFM. F-VCD incorporates a two-stage strategy for both denoising and deconvolution tasks. Overall, I think it is a nice work for high-performance 3D living cell imaging using FLFM. I would recommend its publication in Communication Biology if the following points are addressed.

We thank the Reviewer for recognizing our contribution and the valuable suggestions. We have thoroughly addressed all the questions raised.

1. In terms of hardware, Shu Jia et al [12] have proposed a similarly shaped microlens array to achieve high-resolution FLFM imaging in living cells. But in the experiments of this paper, the proposed DOE seems to play exactly the same role as the microlens array in [12]. Therefore, this DOE does not seem to be of significance. However, I don't think the DOE is very crucial for this paper.

We thank the Reviewer for this great comment. From the results of optical modulation, DOE-based microlens array acts the same role with conventional one in Shu Jia et al [12] because of the similar designed transmittance function. However, DOE is rather easy to be highly customized, and performs excellent fabrication precision (<100 nm), which reduce the difficulty of microlens design, especially for the requirement of large pitch and fill factor in high-resolution FLFM imaging. Therefore, what we want to highlight in this paper is not that DOE has better performance than a conventional microlens array under the same design, but that DOE is a promising and much more low-cost alternative (dozens of times cheaper than customized conventional lens) when building a FLFM system. Furthermore, since DOE has the capability of flexible phase modulation, it allows the combination between FLFM and multi-focus imaging, or extends the depth-of-field of FLFM in the future.

We are sorry that we might lead to misunderstanding in the original manuscript, thus we have modified the following paragraphs in the revised manuscript:

Line 10-12, Paragraph 4, Section "Introduction":

Also, the fabrication of current Fourier microlens arrays is challenging, owing to the requirement of large pitch sizes and small surface curvatures, leading to a high cost.

Line 2-5, Paragraph 1, Sub-section "Principle of F-VCD strategy":

The transmittance function of the DOE was designed to be similar to a previously proposed conventional microlens array¹², but the price was dozens of times cheaper because DOE was much easier to fabricate.

Line 1-5, Paragraph 2, Section "Conclusion and discussion":

In term of light-field encoding, the use of DOE in self-built FLFM dramatically reduces the cost of the system and allows readily-accessible light-field modulation. Since DOE has the capability of

flexible phase modulation, the combination with multi-focus imaging^{33, 34} or enhanced DoF modulation³⁵ might further push the limits of FLFM.

2. In terms of the algorithm, the authors proposed an additional supervised denoising network. A natural question is whether the F-denoise is compatible with both FLFM and LFM data. Can people who only have conventional LFM use F-denoise for image denoising?

We thank the Reviewer for this interesting comment. Generally speaking, the differences between LFM and FLFM are angular sampling rate and individual view's resolution. FLFM yields less captured views for higher spatial resolution while LFM produce dozens of even several hundred views due to the limitation of microlens pitch and the fixed square arrangement style. The F-denoise module is derived from channel-attention mechanism, so changing the number of filters in F-denoise can act the similar denoising ability when applying in LFM.

We have performed an extra simulation experiment of F-denoise on LFM and added following paragraphs in the revised manuscript:

Line 14~18, Paragraph 2, Section "Conclusion and discussion":

It is worth mentioning that, as a view-wise attention denoising strategy, which is independent with the arrangement mode of input views, F-Denoise could also be used to improve the quality of each views and reconstruction of conventional LFM (Supplementary Fig. 10).

We have also added corresponding simulation results in Supplementary Figure 10 (reproduced below) to clarify the universality of F-Denoise in LFM.

Supplementary Figure 10: The compatibility of F-Denoise validated on conventional LFM. With the view-attention denoising module, F-Denoise can be well implemented on conventional LFM. The noisy multi-views of simulated LFM were enhanced by F-Denoise with high SSIM (Right), which also promoted the artifacts suppression and background removal for latter 3D deconvolution reconstruction.

3. In Supplementary Fig. 3, F-VCD shows better performance than VCD and the authors attribute it to the limited-view reconstruction module. An ablation experiment on the module is necessary.

We thank the Reviewer for pointing out the lack of ablation experiment on limited-view reconstruction module. To illustrate the attribution of limited-view reconstruction module, we conduct a comparison between conventional VCD and F-VCD's reconstruction module in **Supplementary Figure 5. We have added the following descriptions to emphasize the contribution of reconstruction module:**

Line 13~15, Paragraph 1, Sub-section "High-resolution reconstruction of fixed cells using F-VCD": Besides, compared with original VCD, F-Reconstruction shows higher spatial resolution and contrast by the extra multi-scale features extraction module (Supplementary Fig. 5).

The reproduced figure:

Supplementary Figure 5: Comparison between VCD and F-Reconstruction. **a** one of the captured triple views. **b** MIP of "F-Denoise + VCD" reconstruction from noisy views. **c** MIP of "F-Denoise + F-Reconstruction" reconstruction from noisy views, which shows higher resolution and contrast than conventional VCD indicated by the plots of ROI and measured K_c . Scale bar: 2 μm

4. In the legend of Fig. 1c, the authors said "real-time inference". But I don't seem to find any relevant description of the inference time of the two networks.

We thank the Reviewer for this constructive question. We have conducted the computational cost comparison between deconvolution and F-VCD (as the following figure shows). Due to the iterative computation approach of deconvolution, the number of iterations is related to reconstruction quality.

Here, we choose two different iteration numbers (10 and 30) as the typical values. For F-VCD, without the need of iteration, it can reconstruct a 3D volume via one inference during 77 ms, providing two orders of magnitude acceleration and yielding 9 folds of voxel counts compared with deconvolution-based approach. **We have added this results in Supplementary Figure 2 and corresponding description in the Section “Principle of F-VCD strategy”.**

The added descriptions:

Line 4~6, Paragraph 3, Section “Principle of F-VCD strategy”: Compared with the low-speed traditional deconvolution-based reconstruction method, F-VCD achieves hundreds of times faster reconstruction (4~13 s vs 77 ms, Supplementary Fig. 2).

Line 11~14, Paragraph 2, Section “Conclusion and discussion”:

Besides, with the fast inference ability of network, F-VCD achieves hundreds of volumetric reconstruction acceleration compared with deconvolution (77 ms vs 4~13 s). This computation time could be even further reduced with multi-GPU parallel computing or model acceleration³⁶ in the future work.

[36] Li, X., Li, Y., Zhou, Y. et al. Real-time denoising enables high-sensitivity fluorescence time-lapse imaging beyond the shot-noise limit. *Nat Biotechnol* 41, 282–292 (2023). <https://doi.org/10.1038/s41587-022-01450-8>

The reproduced figure:

Supplementary Figure 2: The computational cost comparison between deconvolution and F-VCD. DL-based F-VCD achieved two-orders of magnitude acceleration and produced 9 times number of voxels compared with deconvolution. The speed comparison was tested on RTX 3090.

5. Several limitations should be involved in the discussion section, e.g., the shallow depth of field,

inapplicability to in vivo imaging, etc.

We thank the Reviewer for this suggestion. **The following paragraph about the discussion on the limited DOF and future possible improvements has been included in the revised manuscript:**

Line 3~8, Paragraph 3, Section “Conclusion and discussion”:

We believe our proposed imaging strategy would be a valuable and readily-accessible tool for structural studies of cell biology. Here, we only demonstrated the capability of F-VCD on living cell imaging under the high-magnification-imaging hardware setting, which led to a limited DOF (~7 μm). We anticipate to expand the utility of F-VCD to other biological research requiring fast and clear observations, like *in vivo* imaging, by designing suitable hardware settings.

6. *It is recommended that the relevant code be made publicly available rather than requested from the authors so that the broad readers of Communication Biology can easily verify it.*

We thank the Reviewer for pointing out the lack of open source code. **We have uploaded the package containing F-VCD code and example data on GitHub (<https://github.com/feilab-hust/F-VCD>). To be easy-to-use, a manual is provided to illustrate the data processing and networking training/validation.**

7. *Missing reference for other learning-based LFM methods, such as:*

Lu, Z., Liu, Y., Jin, M. et al. Virtual-scanning light-field microscopy for robust snapshot high-resolution volumetric imaging. *Nat Methods* 20, 735–746 (2023). <https://doi.org/10.1038/s41592-023-01839-6>

We thank the Reviewer for pointing out the lack of further review about learning-based LFM methods. **We have added two new citations (ref. 25 & 26) in the revised manuscript:**

Line 1~3, Paragraph 5, Section “Introduction”:

Deep neural networks (DNNs) have emerged as a powerful tool to overcome the limitations imposed by photon budgets for their strong fitting ability and their capacity to incorporate abundant prior knowledge¹⁹⁻²⁶.

25. Lu, Z. et al. "Virtual-scanning light-field microscopy for robust snapshot high-resolution volumetric imaging". *Nat. Methods* 20, 735-746 (2023).

26. Vizcaíno, J.P. et al. "Learning to Reconstruct Confocal Microscopy Stacks From Single Light Field Images". *IEEE Transactions on Computational Imaging* 7, 775-788 (2021).

Reviewer #2:

In this article, the authors combine a two-stage view-channel-depth (VCD) deep learning reconstruction strategy with a Fourier light field microscope based on a customized diffractive optical element to achieve a fast 3D super-resolution reconstruction of intracellular dynamics

measured by a single diffractive limited two-dimensional light field. This Fourier light field imaging method was claimed to improve 3D imaging of intracellular dynamics at video rate (50 volume/second), with high spatiotemporal resolution of $\sim 180\text{ nm} \times 180\text{ nm} \times 400\text{ nm}$ and strong anti-noise capability, and can well reconstruct light field images with a low signal-to-noise ratio (SNR). The authors demonstrated 3D imaging of mitochondrial fission and fusion in the cells. The research has an innovative basis and the results are satisfying. However, there are some issues to be solved in order to be published.

We thank the Reviewer for thoroughly evaluating our work for appreciating our contribution and the constructive suggestions. We have addressed all comments raised.

1. *“We specifically designed two network modules to progressively denoise the corrupted Fourier LFs and reconstruct the denoised LFs. The first module is a view correlated denoise module, termed ‘F-denoise’ module, which converses the noised views into clean ones. .. The second module of F-VCD, functionalized as ‘Fourier light-field reconstruction’, is a ‘limited-view to 3D depth’ transformation module to reconstruct 3D volumes...”. The authors should clearly explain the purpose of doing this, i.e., what is the necessity and significance of the noise reduction image obtained in the first stage reconstruction before the 3D reconstruction in the second stage? In my opinion, the 3D reconstruction already includes the denoising result, why not merge it and get 3D volumes in one step? The reported deep networks have the ability to do this.*

We thank the Reviewer for pointing out the lack of clarity why we design a two-stage framework instead a one-stage one. As the Reviewer mentioned, there are one-step network those can solve the inverse problem under noise influence, like Real ESRGAN¹, which is used to enhance the natural images with different degradation process. However, different from macron photography, fluorescent imaging yield severe noisy images because of the less captured photons ($\sim 10^2$ per pixel for fluorescent microscopy imaging while $\sim 10^5$ per pixel for photography²). For the application of living cell imaging (sensitive to phototoxicity), this severe noise and extra fluorescent background will have negative influence on the network convergence and cause resolution decay and undesirable artifacts during the image upsampling process. Besides, the “pre-denoising strategy” has been also adopted in many current deep learning networks which used a sub-network to enhance the SNR of captured images, like SFSRM³, 3D RCAN⁴, rDL⁵, etc. In essence, this introduction of immediate supervised data is used to narrow the “domain gap” between the noisy data and high-resolution GT according to decompose this reconstruction ill-problem and reduce the SNR requirement for cell imaging.

[1] Morris, P.A., Aspden, R.S., Bell, J.E., Boyd, R.W. & Padgett, M.J. "Imaging with a small number of photons". Nat. Commun. 6, 5913 (2015).

[2] Wang, X., Xie, L., Dong, C. & Shan, Y. in Proceedings of the IEEE/CVF international conference on computer vision 1905-1914 (2021).

[3] Chen, R., Tang, X., Zhao, Y. et al. Single-frame deep-learning super-resolution microscopy for intracellular dynamics imaging. Nat Commun 14, 2854 (2023). <https://doi.org/10.1038/s41467-023-38452->

[4] Chen, J., Sasaki, H., Lai, H. et al. Three-dimensional residual channel attention networks denoise and sharpen fluorescence microscopy image volumes. Nat Methods 18, 678–687 (2021). <https://doi.org/10.1038/s41592-021-01155-x>

[5] Qiao, C., Li, D., Liu, Y. et al. Rationalized deep learning super-resolution microscopy for sustained live imaging of rapid subcellular processes. *Nat Biotechnol* 41, 367–377 (2023). <https://doi.org/10.1038/s41587-022-01471-3>

First, we have added further explanations to clarify the rationale of this network design as follow:

Line 6~9, Paragraph 6, Section “Introduction”:

F-VCD decomposes complex FLFM reconstruction problem into two procedures: view-correlated denoising and limited-view 3D reconstruction, which narrows the gap between the corrupted 2D LFs and high-quality 3D stacks

Second, to illustrate the attribution of denoising module, we have made an ablation study to illustrated the resolution enhancement and artifacts removal brought by this progressive network design as shown in Supplementary Fig 4 (reproduced below) and modified the corresponding the following description in the manuscript:

Line 10~13, Paragraph 1, Sub-section “High-resolution reconstruction of fixed cells using F-VCD”:
Together with the limited-view reconstruction module (F-Reconstruction), this progressive optimization strategy achieved higher 3D resolution, fidelity, and fewer artifacts than one-stage reconstruction network (Supplementary Fig. 4).

The reproduced figure:

Supplementary Figure 4: Comparison of the one-step network (F-Reconstruction) and a two-stage one (F-VCD) on 3D FLFM reconstruction of living mitochondrial. F-VCD reconstruction shows improved resolution and reduced artifacts with the extra F-Denoising module.

2. “As shown in Fig. 2a, ... In contrast, F-VCD remarkably enhanced the spatial resolution of FLFM ...”. Here I did not see the resolution quantification in Fig. 2. I think the authors meant that the F-VCD remarkably enhanced the image contrast which also corresponds to clear detailed structure and is easy to confused with resolution. The spatial resolution was measured and presented

in the next figures but not here.

We thank the Reviewer for this good suggestion. Fig.2 (a) was used to illustrate the higher structure similarity and lower errors of F-VCD compared with other deconvolution-based approaches. **We have modified and added the descriptions about reconstruction quality improvement by F-VCD as follow:**

Line 8~10, Paragraph 1, Sub-section “F-VCD enables high-fidelity reconstruction under various SNR”: In contrast, F-VCD remarkably enhanced the reconstruction quality of FLFM, with which the membrane structure of the mitochondria was clearly reconstructed with higher contrast in both x-y and x-z planes.

In addition, we also revised Fig 2 (a) (reproduced below) to highlight the finer structure details and higher contrast offered by F-VCD.

Fig. 2 Comparative performances of FLM deconvolution, SRRF-deconvolution, and F-VCD on the reconstruction of Mitochondria. a Comparison of 3D reconstruction from deconvolution-based algorithms and F-VCD. The white area denoted the higher contrast and structural details fidelity of F-VCD results compared with deconvolution-based ones. SSIM and NRMSE metrics were calculated to quantitatively evaluate the reconstruction accuracy of F-VCD. **b** The performance of F-VCD under different SNR. Error maps were calculated between ground truths and different F-VCD results.

3. The authors mentioned the resolution improvements several times, but there is no statistical analysis of this, only some subjective selections of a profile across the mitochondria (Figs. 3 and 4). The authors should perform more PSF or profile measurements to verify the resolution improvement, and analyze their statistical parameters, such as mean, variance, p-value, etc.

We thank the Reviewer for pointing out the lack of statistics about the resolution quantification. **We have added corresponding statistical results in Fig. 3 and 4 (reproduced below).**

Fig. 3 Resolution enhancement on fixed mitochondria using F-VCD. **a** Top: raw FLFM views with noises; Bottom: F-VCD reconstruction. **b** magnified views of mitochondria (ROI 1 and 2 marked with white boxes in (a) and their corresponding 3D reconstructions by conventional deconvolution, denoised deconvolution and F-VCD, respectively. The comparative frequency spectrums indicate that F-VCD reconstruct mitochondria with highest spatial resolution. **c** Resolution quantification from frequency analysis on fixed mitochondria. Lateral (left) and axial (right) values are shown for deconvolution reconstruction (Lateral: 442.3 ± 29.2 nm, Axial: 625 ± 54.6 nm, respectively; p-value: 3.4016×10^{-11}) and F-VCD reconstruction (Lateral: 181.3 ± 10.0 nm, Axial: 414 ± 21.6 nm, respectively; p-value: 1.152×10^{-7}).

Fig. 4 3D imaging of fixed endoplasmic reticulum with F-VCD. **a** Comparison of deconvolution-based reconstruction and F-VCD reconstruction. The magnified images of 2 selected ROIs show the significant quality difference between two reconstruction approaches. **b** The y-z section plane of deconvolution results and F-VCD results along the white dash line in (a). The frequency spectrums of the axial planes show the spatial resolution enhancement offered by F-VCD. The white rectangular region denotes F-VCD outperforms in artifacts suppression. Right: The statistics results of resolution quantification of deconvolution approach (lateral: 509.0 ± 67.5 nm, axial: 704.6 ± 70.2 nm; p-value: 2.82×10^{-8}) and F-VCD (lateral: 202.0 ± 23.4 nm, axial: 480.9 ± 37.2 nm; p-value: 9.99×10^{-4}).

4. *The network reconstruction speed is an important metric because we do not want the reconstruction speed to be so slow exceeding the acquisition time of high-SNR images. The author should provide additional information in this regard.*

We thank the Reviewer for this constructive question. We have conducted the computational cost comparison between deconvolution and F-VCD (as the following figure shows). Due to the iterative computation approach of deconvolution, the number of iterations is related to reconstruction quality. Here, we choose two different iteration numbers (10 and 30) as the typical values. For F-VCD, without the need of iteration, it can reconstruct a 3D volume via one inference during 77 ms, providing two orders of magnitude acceleration and yielding 9 folds of voxel counts compared with deconvolution-based approach. **We have added this results in Supplementary Figure 2 and corresponding description in the Section “Principle of F-VCD strategy”.**

The added descriptions:

Line 4~7, Paragraph 3, Section “Principle of F-VCD strategy”: Compared with the low-speed traditional deconvolution-based reconstruction method, F-VCD achieves hundreds of times faster reconstruction (4~13 s vs 77 ms, Supplementary Fig. 2).

The reproduced figure:

Supplementary Figure 2: The computational cost comparison between deconvolution and F-VCD. DL-based F-VCD achieved two-orders of magnitude acceleration and produced 9 times number of voxels compared with deconvolution. The speed comparison was tested on RTX 3090.

5. The annotation and text in the figures are too small to read (especially Figures 1 and 5). The authors should realize that this is very important for reviews' and readers' reading experience. The invisible plots affect our judgment of the reliability of the results.

We appreciate this Reviewer's helpful question. We have changed the annotation and font size in Fig. 1 and 5 (reproduced below).

Fig. 1 The principle of F-VCD.

Fig. 5 Imaging and analyzing mitochondria dynamics in living cell using F-VCD enhanced FLM.

6. The authors claimed to achieve the video rate in 3D imaging of intracellular dynamics at high spatiotemporal resolution, but no time-related results were given, nor did they supplement the video. We appreciate this Reviewer's helpful question. **We have uploaded two videos about long-term living cell imaging (Supplementary Video 1) and dynamic events (Supplementary Video 2), respectively.**

7. A SNR of 1.51 is not dramatically low. The SNR calculation adopted in this work is an alternative definition of SNR is as the reciprocal of the coefficient of variation, i.e., the ratio of mean to standard deviation of a signal or measurement. Notice that such an alternative definition is only useful for variables that are always non-negative, and it is only an approximation. Since GT image is given, SNR can be more precisely calculated as the ratio of the power of signal to the power of background noise. This can be obtained by computing the ratio of summed squared magnitude of the signal to that of the noise (GT - signal). Broadly, it is defined in decibels.

We appreciate this Reviewer's helpful question. The SNR calculation used in this manuscript referenced the commonly-used definition in fluorescent imaging especially when no ground truth data were provided¹⁻³. **To get more accurate metric on simulation data, we have recalculated the SNR referred to the following formula as the Reviewer mentioned:**

$$SNR = 10 \cdot \lg\left(\frac{\sum_{x=0}^{M-1} \sum_{y=0}^{N-1} [f(x, y) - \bar{f}]^2}{\sum_{x=0}^{M-1} \sum_{y=0}^{N-1} [\hat{f}(x, y) - f(x, y)]^2}\right)$$

$$\bar{f} = \frac{1}{MN} \sum_{x=0}^{M-1} \sum_{y=0}^{N-1} f(x, y)$$

, where $f(x, y)$ is the signal distribution and $\hat{f}(x, y)$ is the noisy image. The results were included in the reproduced Fig 2 (also reproduced below) and the above formula was added in Part “Quantification of SNR”, Section ‘Methods’.

[1] Wang, Z. et al. Real-time volumetric reconstruction of biological dynamics with light-field microscopy and deep learning. Nat. Methods 18, 551–556 (2021).
 [2] Zhao, Y., Zhang, M., Zhang, W. et al. Isotropic super-resolution light-sheet microscopy of dynamic intracellular structures at subsecond timescales. Nat Methods 19, 359–369 (2022).
 [3] Chen, J., Sasaki, H., Lai, H. et al. Three-dimensional residual channel attention networks denoise and sharpen fluorescence microscopy image volumes. Nat Methods 18, 678–687 (2021).

Reviewer #3

The authors present a technique to reduce the effect of limited photon budget on the reconstruction quality obtained using Fourier Light Field Microscopy, by employing a deep learning based reconstruction and incorporating a diffractive optical element in place of the commonly used microlens array. In particular, the authors show that it is sufficient to form three light-field projections to generate a 3D reconstruction of the sample using this technique.

Although the approach is interesting, publication in any form is premature at this point. I summarize my main comments below.

We thank the Reviewer for recognizing our contribution and the valuable suggestions. We have thoroughly addressed all the comments raised.

1. *The paper is riddled with spelling and grammatical mistakes. The paper needs to be properly proof-read before resubmission to any journal.*

We appreciate the Reviewer’s good suggestion. **We have thoroughly polished the manuscript and corrected grammatical&&spelling mistakes.**

2. The figure captions need to be more descriptive - it is very difficult to understand the figures at present

We thank the Reviewer's valuable suggestion. **We have added more descriptions in the revised manuscript, specifically as follow:**

The work flow of F-VCD in the Figure 1's caption:

b The pipeline of F-VCD construction, including data generation (the upper row) and F-VCD training (the bottom row). In data generation branch, via simulation, high-resolution stacks are first transformed to light-field views, and then these views are degraded by noise and background quantified from experiments to yield "Noisy LF views". In F-VCD training branch, these noisy views are sent to "F-Denoise" network to suppress the severe noise. Finally, with the F-Reconstruction network, high-resolution 3D results can be produced. Among the network inference, each network is optimized by corresponding loss function (denoise loss for F-Denoise, reconstruction loss for F-Reconstruction)

Detailed description of results in Figure 2's caption:

a Comparison between deconvolution-based reconstruction algorithms and F-VCD reconstruction. The white arrowheads indicate the higher contrast and fidelity of F-VCD. SSIM and NRMSE metrics were calculated to quantitatively evaluate the reconstruction accuracy of F-VCD. **b** The performance of F-VCD under different SNR. Error maps were calculated between ground truths and different F-VCD results.

The description of statistical results in Figure 3's caption:

c Resolution quantification from frequency analysis on fixed mitochondria. Lateral and axial values are shown for deconvolution reconstruction (Lateral: 442.3 ± 29.2 nm, Axial: 625 ± 54.6 nm, respectively; p-value: 3.4016×10^{-11}) and F-VCD reconstruction (Lateral: 181.3 ± 10.0 nm, Axial: 414 ± 21.6 nm, respectively; p-value: 1.152×10^{-7}).

The description of statics results in Figure 4's caption:

b The y-z section plane of deconvolution results and F-VCD results along the white dash line in (a). The frequency spectrums of the axial planes show the spatial resolution enhancement offered by F-VCD. The white frames indicate that F-VCD outperforms in artifacts suppression. The box plots show the statistics results of resolution quantification of deconvolution approach (lateral: 509.0 ± 67.5 nm, axial: 704.6 ± 70.2 nm; p-value: 2.82×10^{-8}) and F-VCD (lateral: 202.0 ± 23.4 nm, axial: 480.9 ± 37.2 nm; p-value: 9.99×10^{-4}).

3. On the same note, figure 1 is very confusing, in particular 1b.

We thank the Reviewer for this helpful question. **We have added more annotations in Figure 1 and corresponding captions. The modified figure is reproduced below.**

Fig. 1 The principle of F-VCD. **a** The optical setup of the FLM system. The inset shows the phase structure of the DOE and its equivalent MLA. **b** The pipeline of F-VCD construction, including data generation (the upper row) and F-VCD training (the bottom row). In data generation branch, high-resolution stacks are first transformed to light-field views, and then these views are degraded by noise and background quantified from experiments to yield “Noisy LF views”. In F-VCD training branch, these noisy views are sent to “F-Denoise” network to suppress the severe noise. Finally, with the F-Reconstruction network, high-resolution 3D results can be produced. Among the network inference, each network is optimized by corresponding loss function (denoise loss for F-Denoise, reconstruction loss for F-Reconstruction). **c** Real-time inference of 3D images from the recorded 2D light-field images by a trained F-VCD model.

4. The text on the plots in figure 5 is impossible to read.

We appreciate this Reviewer’s helpful question. We have made the font larger in Figure 5 (reproduced below).

Fig. 5 Imaging and analyzing mitochondrial dynamics in living cells using F-VCD enhanced FLM.

These issues make it difficult to assess the scientific quality of the paper, and needs to be fixed before a proper review can be made. Nonetheless, I have some questions also on the scientific side: We appreciate the Review's constructive suggestions. We have thoroughly addressed all the comments raised.

5. The approach is motivated by the need for generating high-quality reconstructions with low photon budgets to enable live-cell imaging without phototoxicity. However, this aspect is in my view not properly explored. In particular, I miss a quantitative comparison of the number of photons/volumes needed to perform comparative reconstructions in various imaging modalities.

We appreciate the Review's excellent suggestion. To demonstrate the low phototoxicity of F-VCD-based FLM, we compared WF and F-VCD imaging strategy on 3D long-term observation. As expected, F-VCD strategy could visualize much more volumes than scanning-based WF with high enhanced resolution and higher structure contrast. **We have added the description about the outperformance of F-VCD strategy in the manuscript:**

Line 11~15, Paragraph 1, Sub-section "4D visualization and quantitative analysis of mitochondrial dynamics a living cell":

In sharp contrast, F-VCD maintained the resolutions of reconstructed images across ~5000 times of 3D observation (Fig. 5b, Supplementary Video 1), indicating the better robustness of F-VCD under low photon budget, compared with scanning-based imaging modalities (less than 50 volumes, Supplementary Fig. 7).

and included the results in Supplementary Figure 7 (reproduced below).

Supplementary Figure 7: The robustness of F-VCD under low photon budget assayed by long-term living cell imaging. **a** Wide-field scanning imaging for the living U2OS cell with lysosome. Due to the low efficiency of photon-utilization of WF, the repeat excitation by sequential scanning produce dramatical photobleaching denoted by the intensity decrease of the 50th Vol and frequency bandwidth reduction. Scale bar: 2 μm . **b** F-VCD reconstruction from captured 2D FLFM images. With the denoising ability and resolution enhancement of our network, the reconstruction results perform detailed structure reconstruction (the ring-like structure of lysosome outer membrane) until the 3500th volume. **c** The normalized intensity plots of WF and FLFM captures, which shows the snapshot volumetric imaging of FLFM outperforms the scanning-based approaches in long-term imaging due to its high photon efficiency.

6. It is unclear, which of the improvements are due to the diffractive optical element, and which are due to the deep learning algorithm. This needs to be properly explained

We thank the Reviewer for this valuable question. The improvements compared with conventional approach are attributed to the noise robustness and resolution enhancement offered by the deep learning algorithm. We designed a two-stage network to solve the subproblems (multi-view denoising and super-resolution 3D reconstruction) decomposed from 3D reconstruction in living cell imaging. DoE adopted in this manuscript was to bypass the high manufactury accuracy requirement for conventional microlens array and reduce the hardware implementations cost. **We have modified the following descriptions about the function of DoE and F-VCD:**

Line 1~5, Paragraph 2, Section “Conclusion and discussion”:

In term of light-field encoding, the use of DOE in self-built our FLMF dramatically reduces the cost of the system and allows readily-accessible light-field modulation. Since DOE has the capability of flexible phase modulation, the combination with multi-focus imaging or enhanced DoF modulation might further push the limits of FLMF.

Line 5~11, Paragraph 2, Section “Conclusion and discussion”:

In term of light-field reconstruction, our F-VCD shows ~2.2-times improved resolution accompanied with ~30% increased reconstruction accuracy, as compared to conventional deconvolution-based FLMF approach. Also, unlike regular VCD that directly predict depth information from corrupted views, F-VCD follow a divide-and-conquer strategy to fulfill the coupled denoise and view-to-depth transformation tasks, thereby showing notably improved performance when dealing with live-cell imaging with various conditions.

7. There is no quantitative validation of the deep neural network performance. This is extremely important in this type of work, seeing as the networks are trained to optimize some metric on a large dataset, which does not necessarily guarantee that the output can be used to draw downstream biological conclusions. Validating the biological conclusions on some test case where conventional imaging techniques can be used would be very valuable.

We thank the Reviewer for this constructive question. We have conducted an *in situ* experiment that imaging the fixed cell with confocal microscopy (IXplore SpinSR, Olympus) and our self-built FLMF microscopy. We used the Squirrel analysis¹ to evaluate the fidelity and accuracy of our proposed F-VCD reconstruction. In addition, we also performed quantitative analysis between F-VCD reconstructions and captured raw views (which is equal to wide-field microscopy with an extended DOF) to evaluate the accuracy of F-VCD on dynamic living cells. **The results are included in Supplementary Figure 9 (reproduced below) and modified the following descriptions in revised manuscript:**

Line 24~29, Paragraph 1, Section “4D visualization and quantitative analysis of mitochondrial dynamics a living cell”:

For the quantitative validation of F-VCD’s performance, we conducted *in situ* comparison between traditional imaging methods (confocal and wide-field imaging) and F-VCD reconstruction both on fixed and living cells, and Squirrel analysis³² was performed. The results showed high structural similarity and few reconstruction errors of F-VCD, indicating that the proposed imaging strategy

could be used to perform biological downstream analysis.

[32] Culley, S., Albrecht, D., Jacobs, C. et al. Quantitative mapping and minimization of super-resolution optical imaging artifacts. *Nat Methods* 15, 263–266 (2018). <https://doi.org/10.1038/nmeth.4605>

The reproduced figure:

Supplementary Figure 9: Quantitative validation of F-VCD reconstruction. **a** The *in situ* imaging comparison between confocal microscopy and F-VCD reconstruction via FLFM on fixed U2OS cell with mitochondria. F-VCD results perform higher contrast and detailed structure. The high RSP (above 0.9) and low RSE (~10) denote F-VCD's comparable accuracy with conventional confocal imaging. **b** Quantitative evaluation of dynamic reconstruction. *Left*: Represent images of captured view and corresponding F-VCD reconstruction during the time window of dynamic events. The red arrows denote the local morphology of mitochondria. *Right*: The statistical results of error evaluation in living cell imaging. The mean RSP and RSE of F-VCD reconstruction show the similar accuracy across the dynamic events.

REVIEWERS' COMMENTS:

Reviewer #1 (Remarks to the Author):

The authors have fully addressed my concerns and I recommend publication of this nice manuscript.

Reviewer #2 (Remarks to the Author):

The author has thoroughly addressed my comments and made appropriate revisions to the manuscript. This includes adding more explanations, experiments, analyses, and discussions to clarify questions raised by the reviewers. Specifically, the author added ablation studies, statistical quantification, computational cost comparison, open source code, etc. to better validate the method and results. The revisions have improved the clarity, completeness, and validity of the manuscript.

Reviewer #3 (Remarks to the Author):

The authors have addressed my concerns from the previous iteration. I suggest publication.

Re: Revision for “Communications Biology” Manuscript “**Video-rate 3D imaging of living cells using Fourier view-channel-depth (F-VCD) light field microscopy**” (# COMMSBIO-23-2128A) authored by Chengqiang Yi, Lanxin Zhu, Jiahao Sun, Zhaofei Wang, Meng Zhang, Fenghe Zhong, Luxin Yan, Jiang Tang, Liang Huang, Yu-Hui Zhang, Dongyu Li (corresponding author) and Peng Fei.

We are highly grateful to the reviewers for their recognition on our work. We have reformatted the manuscript and uploaded required materials referred to the final submission file checklist and editorial requests file.

Sincerely yours,

Dongyu Li,
Associated Professor of School of Optical and Electronic Information,
Huazhong University of Science and Technology; Email: li_dongyu@hust.edu.cn